# Identifying Predictors of Neck Disability in Patients with Cervical Pain Using Machine Learning Algorithms: A Cross-Sectional Correlational Study

**DOI:** 10.3390/jcm13071967

**Published:** 2024-03-28

**Authors:** Ahmed A. Torad, Mohamed M. Ahmed, Omar M. Elabd, Fayiz F. El-Shamy, Ramzi A. Alajam, Wafaa Mahmoud Amin, Bsmah H. Alfaifi, Aliaa M. Elabd

**Affiliations:** 1Basic Science Department, Faculty of Physical Therapy, Kafrelsheik University, Kafrelsheik 33516, Egypt; ahmed_alimohamed@pt.kfs.edu.eg; 2Department of Physical Therapy, Collage of Applied Medical Sciences, Jazan University, Jizan 45142, Saudi Arabia; ralajam@jazanu.edu.sa (R.A.A.); wafaa_770@yahoo.com (W.M.A.); bhjabor@jazanu.edu.sa (B.H.A.); 3Department of Basic Sciences, Faculty of Physical Therapy, Beni-Suef University, Beni-Suef 62521, Egypt; 4Department of Orthopedics and Its Surgery, Faculty of Physical Therapy, Delta University for Science and Technology, Gamasa 35712, Egypt; 3omarel3abd@gmail.com; 5Department of Physical Therapy, Aqaba University of Technology, Aqaba 11191, Jordan; 6Department of Physical Therapy for Women Health, Kafrelsheikh University, Karfelsheikh 33516, Egypt; ff_elshamy@yahoo.com; 7Department of Basic Sciences of Physical Therapy, Faculty of Physical Therapy, Cairo University, Giza 12613, Egypt; 8Department of Basic Sciences, Faculty of Physical Therapy, Benha University, Benha 13511, Egypt; aliaelabd88@gmail.com

**Keywords:** neck pain, neck disability, machine learning algorithms, predictors, chronic pain, cross-sectional study

## Abstract

(1) **Background:** Neck pain intensity, psychosocial factors, and physical function have been identified as potential predictors of neck disability. Machine learning algorithms have shown promise in classifying patients based on their neck disability status. So, the current study was conducted to identify predictors of neck disability in patients with neck pain based on clinical findings using machine learning algorithms. (2) **Methods:** Ninety participants with chronic neck pain took part in the study. Demographic characteristics in addition to neck pain intensity, the neck disability index, cervical spine contour, and surface electromyographic characteristics of the axioscapular muscles were measured. Participants were categorised into high disability and low disability groups based on the median value (22.2) of their neck disability index scores. Several regression and classification machine learning models were trained and assessed using a 10-fold cross-validation method; also, MANCOVA was used to compare between the two groups. (3) **Results:** The multilayer perceptron (MLP) revealed the highest adjusted R2 of 0.768, while linear discriminate analysis showed the highest receiver characteristic operator (ROC) area under the curve of 0.91. Pain intensity was the most important feature in both models with the highest effect size of 0.568 with *p* < 0.001. (4) **Conclusions:** The study findings provide valuable insights into pain as the most important predictor of neck disability in patients with cervical pain. Tailoring interventions based on pain can improve patient outcomes and potentially prevent or reduce neck disability.

## 1. Introduction

Neck pain, a prevalent musculoskeletal disorder affecting a significant portion of the population, poses a substantial burden on individuals’ well-being and quality of life. The estimated prevalence rate was 27.0 per 1000 population in 2019 [1]. Neck pain represents a widespread concern that warrants detailed investigation. While many instances of neck pain are transient and self-limiting, a subset of individuals experience chronic and debilitating symptoms that have profound impacts on their daily activities, posture, and overall quality of life [2]. Addressing these chronic cases of neck pain requires a comprehensive understanding of the factors that contribute to neck disability and hinder effective interventions.

Cervical pain is not only prevalent but also clinically significant, as it may contribute to cervicogenic headaches [3] and influence changes in the masticatory muscles [4], thereby highlighting the clinical importance of the cervical region. The quest to improve outcomes for individuals suffering from chronic neck pain is underscored by the need to identify key prognostic variables that can guide clinical decision-making and intervention strategies [5]. These variables not only assist in predicting the course of neck pain but also enable the tailoring of targeted treatments that mitigate or prevent neck disability.

In recent years, machine learning algorithms have emerged as a promising avenue for tackling the complexity of neck pain and disability. These algorithms offer a data-driven approach to analyse objective measures of neck function and related factors, facilitating the identification of distinct levels of neck disability among patients. Recent studies have showcased the potential of machine learning algorithms in achieving high levels of accuracy in classifying patients based on their neck disability status. Nevertheless, the emphasis of these studies has primarily been on achieving accurate classification, and limited attention has been devoted to the underlying predictors of neck disability as identified by machine learning algorithms [6]. This point is not in the literature, but machine learning has been used a lot to identify risk factors for many issues like anxiety [7], energy and fatigue [8], cardiovascular diseases [9], and breast cancer [10].

This study seeks to bridge this gap in research by employing a range of machine learning algorithms to construct both classification and regression models. These models will incorporate various demographic, clinical, and physical function measures as potential predictors of neck disability. The central objective is to uncover the key predictors of neck disability through rigorous feature selection methods, shedding light on the intricate mechanisms that contribute to neck pain outcomes. By providing insights into these predictors, this study aims to empower clinicians with valuable information to develop targeted interventions that effectively alleviate or prevent neck disability in patients grappling with cervical pain.

## 2. Materials and Methods

### 2.1. Population

In accordance with the 1964 Helsinki Affirmation and subsequent amendments, a cross-sectional correlational study was carried out to explore the correlations between neck pain intensity, neck disability, cervical contour (the external posterior aspect of the surface of the cervical spine), and surface electromyographic (sEMG) characteristics (muscle activities and fatigue levels) of the axioscapular muscles (levator scapula and upper trapezius). The study received approval from KafrElshikh University (P.T/BAS/5/2023/49) by the Faculty of Physical Therapy’s research ethics committee, which was also registered in the clinical trial registry under the number NCT05291377. Each participant gave their written consent before being included in the study, which was voluntary.

The study’s focus was on patients with chronic neck pain who were between the ages of 20 and 40. An orthopaedic surgeon gave them all chronic neck pain diagnoses. The initial survey was completed by those who replied to the open invitation. The first author, who had more than ten years of expertise, conducted standardised physical examinations and evaluated eligibility criteria [11]. Ninety patients who met the eligibility criteria and gave their written consent took part in the trial after being fully informed of its procedures and risks.

Chronic neck pain with at least three months of lasting symptoms and mild neck illness were the inclusion criteria [12]. According to the median value (22.2) of the NDI, participants were categorised into two groups: (A) high disability (NDI > 22.2) and (B) low disability (NDI ≤ 22.2). The following conditions were excluded: neck pain currently receiving physical therapy or medical treatment; cervical spine congenital disorders, disc prolapse, contracture, or surgery; pathologies affecting the cervical spine such as systematic inflammatory diseases and skin conditions; and visual or auditory issues. Also, patients with mental disorders were excluded from the study.

Neck pain intensity, functional neck disability, cervical spine curvature, and axioscapular muscle (levator scapula and upper trapezius) sEMG parameters (muscle activity and fatigue) were measured. Anti-inflammatory medications were to be avoided for 72 h before the investigation. Outcome measures and demographic and clinical data were obtained by an examiner who was unaware of the patients’ allocation while the participants finished the self-report questionnaires.

### 2.2. Procedures

The severity of the neck pain was assessed by the visual analogue scale (VAS). Each participant marked the place on the line that mirrored their pain to indicate the severity of their pain. By measuring from the left line’s end to the specified point, the score was calculated. This is an efficient and simple measurement tool with proven validity and reliability [13,14].

The neck disability index (NDI), a questionnaire of 10 questions, was used to assess functional neck disability. The severity of each question, ranging from 0 (no disability) to 5 (complete disability), was marked by each participant. The total score was calculated by adding all the marks and dividing them by 50, or 45 if one component was missed. The validity and reliability of the NDI have been established [15,16]. According to the median value (22.2) of the NDI, participants were categorised into two groups: (A) high disability (NDI > 22.2) and (B) low disability (NDI ≤ 22.2).

A flexible ruler (ati, FC-700R, Taiwan) was used to measure the cervical spine curve. It is affordable, transportable, simple to operate, and reliable [17,18,19]. Even though it has very high reliability (0.91) with known MCID, it is not a valid tool (r = 0.15) for the measurement of cervical curvature on X-rays [19]. To calculate the angle, the patient’s elbows were resting on an armrest directly below the acromion as the patient sat in a chair with their feet flat on the floor, and the ruler was firmly pressed against the patient’s upright cervical spine between the occiput and the seventh cervical spinous process. A mathematical equation was used, angle (Q) = arc tan (2b/a) [20], as shown in Figure 1.

The skin was meticulously prepared to ensure low impedance and high-quality signal acquisition. This involved shaving the hair at the electrode placement sites and thoroughly cleansing the skin with cotton and alcohol to remove any oils or debris. This preparation aimed to minimise impedance and ensure the reliability of the sEMG signals recorded during the study. The study was conducted during standard laboratory hours, ensuring consistent environmental conditions for all participants [21].

Electrode placement was carefully executed to ensure accurate and reliable sEMG recordings. The placements were as follows: for the upper trapezius, two cm laterally from the midpoint between the spinous process of C7 and the posterolateral acromion; for the levator scapula, laterally to the spinous process of C3, C4, between the upper trapezius and sternocleidomastoid (Figure 2). A reference electrode was placed over the C7 spinous process. These placements were recommended and used in previous studies [2,22,23], and they are presented in Figure 2. The placements are in accordance with the SENIAM program guidelines, ensuring standardised and reproducible electrode positioning. The electrodes used were silver/silver chloride (Ag/AgCl) surface electrodes with a conductive surface area of 10 mm in diameter, supplied by Noraxon Inc. The maximum acceptable electrode impedance was set below 5 kΩ to ensure signal quality [23].

During the sEMG study, participants were positioned with their back fully supported, feet flat on the floor, and hips and knees flexed at 90 degrees to standardise posture and minimise external influences on muscle activity. The research procedure included measuring maximal voluntary isometric contractions (MVICs) for both muscles of interest, followed by a writing task performed for 15 min to simulate a typical daily activity that exacerbates symptoms. The MVIC measurements were conducted with specific postures for the upper trapezius and levator scapula, and each contraction was held for 7 s, with a 30 s rest between the three repetitions. Isometric shoulder abduction with the arm at 90° abduction and neutral rotation was used for the upper trapezius, and static shoulder elevation against manual resistance over the shoulder while maintaining lateral rotation of the neck to the same side was used for the levator scapula [24].

The MyoSystemTM 1400A (Noraxon Inc., Scottsdale, AZ, USA) device was used to capture the sEMG signals. This system boasts a sampling rate of 1000 Hz, providing high-resolution data. The bandwidth was set between 20 and 450 Hz with a high-pass filter cutoff frequency at 20 Hz and a low-pass filter at 450 Hz. The input impedance of the sEMG system was 10 GΩ, and the common mode rejection ratio was greater than 80 dB at 60 Hz, ensuring high signal quality and minimal interference. The input range was ±2.5 V, and the baseline noise level was minimised to ensure clear signal capture. These specifications underscore the reliability and precision of the sEMG recordings obtained during the study [2,23].

The raw sEMG signals underwent a systematic processing protocol to prepare them for analysis. Initially, full-wave rectification was performed to correct for any systemic bias in the signals. This was followed by filtering according to the specified bandwidth to eliminate any unwanted frequencies. The signals were then normalised to the average of the three MVIC trials for each participant, facilitating the comparison of muscle activation levels across different activities. The normalisation process was crucial for interpreting the muscle’s response to the semi-static load imposed by the writing task. No automatic processing was used; instead, manual filtering and normalisation procedures were applied. The processing and analysis were conducted using the proprietary software provided by Noraxon Inc, ensuring consistency and reliability in the signal processing workflow [24].

Pilot testing with 3 patients per group (high disability/low disability) was conducted to determine the sample size using VAS as the dependant variable with the effect size of 0.53 and power of 80%; the proposed sample size was 90 patients.

Python V.3.8.5 (Python Software Foundation, Wilmington, DE, USA) was used to carry out all initial and primary analyses.

### 2.3. Statistical Analysis

Following the determination of the crucial sEMG parameters, data were gathered from all patients using a data collection form. The collected data were then processed further for training. No form of imputation was utilised because there were no missing values observed. After that, the data were checked for outliers and refined to lessen the impact of extreme values. Refining includes replacing extreme values with values that fall inside a particular percentile of the original distribution. In order to recode values above the 95th percentile to the 95th percentile value, all variables in this study were rescaled.

According to the median value (22.2) of the NDI, participants were categorised into two groups: (A) high disability (NDI > 22.2) and (B) low disability (NDI ≤ 22.2). To determine differences between groups in sex distribution, Chi-squares were used, and for age, height, weight, and BMI T-tests were used. The following analyses were then carried out.

Objective 1: Model Training and Evaluation

For classification models: Participants were categorised into high and low disability groups based on the median value (22.2) of the neck disability index (NDI). Sex distribution differences were determined using Chi-squares, while t-tests were employed for age, height, weight, and BMI comparisons. The dataset was used for training classifiers, employing models such as random forest classifier, MLP classifier, decision tree classifier, K-Neighbours classifier, Gaussian, and linear discriminant analysis. To address overfitting and selection bias, 10-fold cross-validation was applied. The mean and median accuracy, as well as the AUC of the receiver characteristic operator (ROC), were used to evaluate classification models [8].

For regression models: Regression models were employed with the NDI as continuous variables. Models such as K-neighbours regression, decision tree regression, random forest regression, ridge regression, stochastic gradient descent (SGD) regression, MLP regression, SVR, and gradient boosting regression were utilised. Similar to the classification task, overfitting was mitigated through 10-fold cross-validation. The mean absolute error (MAE) assessed regressor accuracy, while correlation coefficients (R2) evaluated the relationship between expected and self-reported disability scores [7].

To some extent, overfitting and selection bias was avoided by training each model using 10-fold cross-validation. The training set of the models (90%) and the test set (10) were divided at random [25]. The mean and median accuracy and the area under the curve (AUC) of the receiver operator characteristic (ROC) were used to assess classification models. The accuracy of the regressor was evaluated using the mean absolute error (MAE). Between the expected disability score and the self-reported disability score on the regression models, correlation coefficients (R2) were evaluated as well [26].

Objective 2: Feature importance

Not all high-dimensional aspects are equally relevant, and there could be a lot of redundant features that are less significant. Thus, the calculation for the feature importance was conducted depending on the highest model available using its feature importance function or permutation.

Objective 3: Differences in outcome characteristics between groups

Age, weight, height, and BMI were utilised as co-variants in a 2 (high/low disability groups) × 2 (sex) mixed model analysis of covariance (MANCOVA) to investigate differences between the low and high disability groups. Normality assumptions for all variables were tested between the two groups using a combination of histograms and the Shapiro–Wilks test before the primary analysis. Despite the use of exponential, power, arcsine, and logarithmic transformations for variables that were not normally distributed, the histograms and Shapiro–Wilks tests did not show significant differences compared to the original variables. Consequently, due to the lack of non-parametric versions of a MANCOVA, we used a MANCOVA by combining large sample theory. To investigate differences among the independent groups, a full-factorial general linear model was employed, along with a polynomial multivariate contrast. If necessary, Bonferroni’s post hoc pairwise comparison was conducted, and the significance level (α) was set at 0.05. Also, Bonferroni correction was used. It is important to note that all analyses and *p*-values presented in the results have been adjusted accordingly.

## 3. Results

The flowchart of the participants was presented in Figure 3.

Out of the 105 initial subjects, 15 were excluded due to different reasons. Ninety patients met the eligibility criteria and participated in the study. There were 44 participants (males = 9, females = 35) in the low disability group and 46 participants (males = 11, females = 35) in the high disability group (Figure 2). There were no significant differences between the two groups regarding the demographic characteristics; age, weight, height, BMI, and sex (*p* > 0.05) Table 1.

While multi-layer perceptron had a mean of 0.708 (0.645, 0.771) and median of 0.727 with R2 equal to 0.67 and adjusted R2 equal to 0.768 in the time, the absolute error was −3.83, and the squared error equalled −25.163, making it the most precise regression model with the greatest R2 to determine people with low and high disability, as shown in Table 2.

The linear discriminate analysis had an F1 score of 0.83 and ROC AUC of 0.91 in the time, and precision equalled 0.859 and sensitivity equalled 0.839, with the negative likelihood of −0.112 and negative log loss of −0.371, making it the most precise classification model with the highest AUC ROC to determine people with low and high disability, as shown in Table 3.

For both the classifier and regression models, the variable VAS was ranked as the most important feature for the model of disability. This indicates that the VAS (could stands for visual analogue scale) has the highest predictive power in distinguishing between different disability classes. Gender and weight were ranked second and third, suggesting that they are significant factors in predicting disability, as shown in Table 4.

There were statistically significant differences between the two groups regarding all variables, as shown in Table 5.

## 4. Discussion

The primary objective was to investigate the application of machine learning in identifying disability in patients with cervical pain. Participants were categorised based on their scores on the NDI, with those scoring above the median value of 22.2 categorised as having high disability and those scoring ≤ 2.22 categorised as having low disability. Machine learning algorithms were used to predict spine disability in patients with neck pain. Random forest classifiers achieved the highest accuracy for predicting neck disability, with age, gender, and pain intensity being the most important predictors [27,28]. However, this is the first study to use machine learning models to identify disability based on objective findings.

Our results showed that the linear discriminant analysis model was the most accurate classifier, with an F1 score of 0.83 and an ROC AUC of 0.91. The most important features in this model were the VAS, gender, weight, upper trapezius median frequency, age, height, cervical spine surface contour, levator RMS, upper trapezius RMS, BMI, and levator median frequency.

On the other hand, the multi-layer perceptron regression model was the most accurate for determining people with low and high disability, with an R2 value of 0.67 and an adjusted R2 value of 0.768. The most important factors in this model were not explicitly mentioned, but we can infer that they were likely related to the predictors used in the regression analysis (i.e., age, height, weight, and BMI).

The study results revealed that pain level, gender, and weight were the top three predictors for disability in patients with cervical pain. The findings are consistent with previous studies that have shown a strong relationship between pain and disability in patients with musculoskeletal disorders [29,30]. The strong predictive power of pain level in this study underscores the importance of proper pain management as a crucial aspect of the treatment plan for patients with cervical pain.

It is not surprising to have a strong relation between cervical pain and disability, as most items on the neck disability index primarily focus on clinical suffering during functional activities. Further, the relationship may be explained by the fear avoidance hypothesis, which holds that a painful experience may set off a maladaptive cognitive process that results in catastrophic thinking, the re-development of fear, and its subsequent misuse, which causes disability. Furthermore, among neck pain sufferers, self-efficacy and psychological distress may act as moderators in the association between pain and disability [31,32,33].

Furthermore, gender was found to be a significant predictor of disability in this study. This finding is consistent with previous studies that have reported a higher prevalence of disability in women than in men with musculoskeletal pain. Although the underlying mechanisms for this sex difference in disability are not fully understood, it has been suggested that psychosocial and cultural factors may play a role [34]. Another explanation may be related to muscle strength. It is believed that females have less muscular strength than males. Further, biomechanical factors related to the postural changes among males and females may be another contributor [35,36].

Weight was also found to be an important predictor of disability in previous studies. Several studies have revealed a positive correlation between disability and BMI in patients with musculoskeletal pain. The relation between weight and disability may be explained by the additional mechanical stress that excess weight places on the musculoskeletal system, leading to increased pain and disability [37,38]. Our study results contradict basic research from 1993 that states sEMG is a good predictor of patient status and prognosis [39].

The findings of this study have significant clinical implications for the management of patients with cervical pain. The identification of pain level, gender, and weight as predictors of disability can assist clinicians in the early detection of patients at risk of developing disability and enable targeted interventions. Clinicians can focus on implementing pain management strategies and weight loss interventions to reduce disability and enhance the quality of life for individuals with cervical pain. Moreover, the study highlights the potential benefits of incorporating machine learning algorithms into clinical practice, which can enhance the accuracy and efficiency of diagnosis and treatment planning.

However, further research is needed to validate and expand upon the findings of this study. Larger sample sizes, diverse populations, and the inclusion of additional relevant factors should be considered in future studies. These endeavours will help confirm the role of machine learning algorithms in improving the assessment and management of patients with cervical pain, particularly in identifying those at high risk of neck disability and tailoring interventions accordingly.

In revising the limitations section of the study, it is important to acknowledge significant methodological constraints that may have impacted the outcomes. Firstly, the use of the flexicurve to measure the cervical curve externally is highlighted as a key limitation, as its accuracy in reflecting the true measurement of cervical lordosis is questioned [19]. Additionally, the study failed to include precise measurements of the cervical spine posture and curvature, which is a significant oversight given the established relationship between forward head posture, cervical lordosis, and neck pain [40]. This absence of proper biomechanical assessments, such as the cranio-vertebral angle (CVA), may undermine the reliability of the study’s regression model outcomes. Finally, the study neglected to consider crucial socio-economic variables among participants, such as education levels and smoking status, which are known factors affecting pain and disability. This omission could significantly skew the study’s findings, making it essential to address these limitations comprehensively to enhance the validity and applicability of the research.

## 5. Conclusions

This is the first study to use machine learning models to identify disability based on objective findings. This study highlights the strong predictive power of pain level, gender, and weight in relation to disability among patients with cervical pain. These findings are valuable for clinicians as they aid in the identification of patients at risk of developing disability and enable the customisation of interventions to mitigate disability and enhance the quality of life.

Furthermore, our study provides initial evidence supporting the potential utility of machine learning techniques in identifying disability among patients with cervical pain. However, further research is necessary to validate these findings and ascertain the applicability of these models to different populations and pain conditions. Conducting additional investigations can strengthen our understanding of the role of machine learning in identifying disability and improve its implementation in clinical practice.

## Figures and Tables

**Figure 1 jcm-13-01967-f001:**
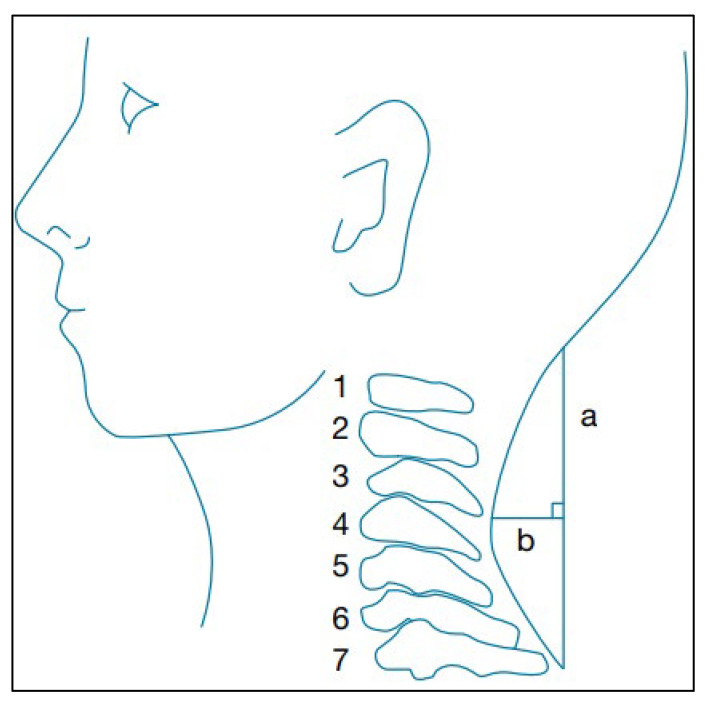
Measurement of cervical curve with flexible ruler. a: Length between the two endpoints of the cervical curve; b: length of the perpendicular line from the midpoint of line a to the curve [20]. 1–7 means seven cervical vertebrae.

**Figure 2 jcm-13-01967-f002:**
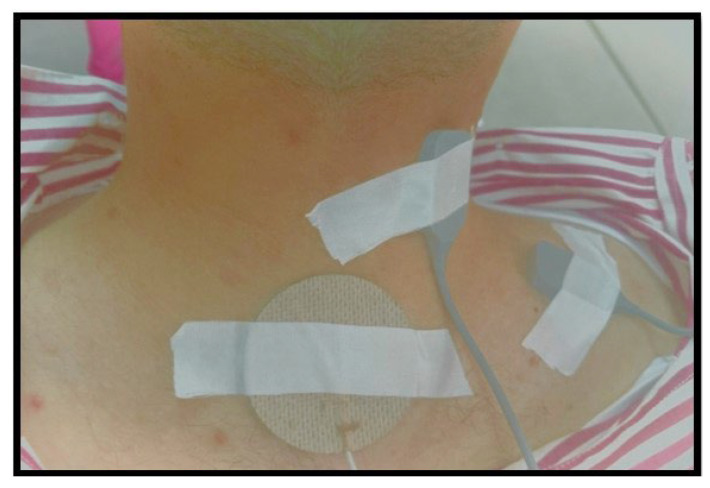
Electrode locations adapted from [2].

**Figure 3 jcm-13-01967-f003:**
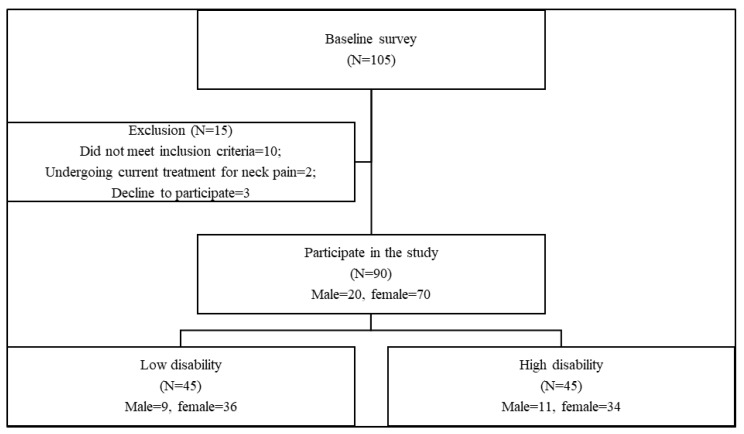
Patient flowchart.

**Table 1 jcm-13-01967-t001:** Participant characteristics.

	Low Disability	High Disability	Sig.
Male/Female *	9:36	11:34	0.695
Age (Years)	26.91 ± 3.99	27.48 ± 3.9	0.495
Height (m)	1.64 ± 0.05	1.64 ± 0.05	0.721
Weight (KG)	74.66 ± 9.22	73.93 ± 9.91	0.875
BMI (KG/H^2^)	28.06 ± 3.18	27.92 ± 3.44	0.842

*: Nominal variable.

**Table 2 jcm-13-01967-t002:** Regressor models across different disability groups.

Model	Mean	Lower CI	Upper CI	Min	Q1	Q2	Q3	Max
MLP	0.708	0.645	0.771	0.272	0.646	0.727	0.827	0.961
RF	0.583	0.528	0.638	0.243	0.49	0.576	0.71	0.848
KN	0.405	0.299	0.512	−0.324	0.269	0.427	0.599	0.841
DT	0.361	0.257	0.466	−0.463	0.255	0.369	0.559	0.843
Ridge	0.336	0.288	0.384	0.061	0.269	0.32	0.44	0.541
SVR	0.037	0.024	0.049	−0.049	0.03	0.045	0.062	0.08
SGD	−0.003	−0.021	0.015	−0.115	−0.035	0	0.028	0.085
GB	−0.148	−0.369	0.072	−1.302	−0.526	−0.008	0.364	0.603

Decision tree: DT; gradient boosting: GB; K-Neighbours: KN; multi-layer perceptron: MLP; random forest: RF; stochastic gradient descent: SGD; support vector regression: SVR.

**Table 3 jcm-13-01967-t003:** Classifier models across disability groups.

Model	Mean	Lower CI	Upper CI	Min	Q1	Q2	Q3	Max
LDA	0.841	0.798	0.884	0.556	0.778	0.889	0.889	1
GNB	0.793	0.734	0.851	0.333	0.778	0.833	0.889	1
RF	0.789	0.73	0.848	0.444	0.667	0.833	0.889	1
MLP	0.759	0.718	0.8	0.444	0.667	0.778	0.778	1
DT	0.748	0.691	0.806	0.444	0.667	0.778	0.861	1
KN	0.711	0.654	0.768	0.333	0.667	0.722	0.778	1

Decision tree: DT; Gaussian naive bayes: GNB; K-Neighbours: KN; linear discriminant analysis: LDA; multi-layer perceptron: MLP; random forest: RF.

**Table 4 jcm-13-01967-t004:** The ranked features for best classifier and regressor models of disability.

Rank	Classifier Variable	Regressor Variable
1	VAS	VAS
2	Gender *	Gender *
3	Weight	Weight
4	UT_MDF	LV_MDF
5	Age	LV_NOR_RMS
6	Height	UT_MDF
7	Surface contour of flexicurve	UT_NOR_RMS
8	LV_NOR_RMS	Height
9	UT_NOR_RMS	Surface contour of flexicurve
10	BMI	BMI
11	LV_MDF	Age

*: nominal variable; LV: levator scapulae; MDF: median frequency; NOR: normalised; RMS: Root Mean Square; UT: upper trapezius; VAS: visual analogue scale.

**Table 5 jcm-13-01967-t005:** Post hoc significant differences between groups.

Variable	Low Disability	High Disability	Sig.	Partial Eta Squared
VAS	3.14 ± 1.23	5.87 ± 1.41	<0.001 *	0.568
Surface contour of flexicurve	32.48 ± 4.16	23.8 ± 6.63	<0.001 *	0.410
LV_NOR_RMS	9.06 ± 9.16	15.07 ± 10.31	0.002 *	0.105
UT_MDF	75.97 ± 18.7	61.64 ± 16.38	<0.001 *	0.172
UT_NOR_RMS	4.98 ± 3.8	10.74 ± 6	<0.001 *	0.298
LV_MDF	71.78 ± 15.45	59.38 ± 13.24	<0.001 *	0.215

LV: levator scapulae; MDF: median frequency; NOR: normalised; RMS: Root Mean Square; UT: upper trapezius; VAS: visual analogue scale.

## Data Availability

Data are available upon request from corresponding author.

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
