# Peer review of "Identifying Predictors of Neck Disability in Patients with Cervical Pain Using Machine Learning Algorithms: A Cross-Sectional Correlational Study"

_jcm, 2024, doi:10.3390/jcm13071967_

Round 1
Reviewer 1 Report
Comments and Suggestions for Authors
Dear Authors, I have the following comments:
1. L49-51 - The authors refer to epidemiological data that is not up to date - update. In my opinion, this is the most up-to-date report - 10.1186/s12891-021-04957-4
2. L51 – ‘’ neck pain represents’’ - Please start a sentence with a capital letter.
3. L57 - Add information that cervical pain is clinically relevant as it may affect the cervicogenic headache DOI: 10.1016/j.msksp.2023.102787 and muscular changes of the masticatory muscles DOI: 10.3390/ijerph19031577 This will highlight the clinical significance of the region.
4. L72 – ‘’ Anxiety [5], Energy’’ - do not use a capital letter in the middle of a sentence.
5. L72 – 73- Once authors have bold quotes other times they are not bold. Standardise the citation style. According to the journal's guidelines, citations should not be in bold. Once authors have bold quotes other times they are not bold. Standardise the citation style. According to the journal's guidelines, citations should not be in bold.
6. L90-93 - This information is not needed. I suggest deleting it.
7. L107-108 –‘’ conducted standardised physical examinations and evaluated eligibility criteria.’’ - Add a citation or opsize this study. The ''standard survey'' may vary depending on the region of the world.
8. Figure 1 - Figure 1 is cropped from the top, correct it.
9. L115 –‘’ (Vernon, 2008) [9]’’ - Standardise the citation style.
10. L115-118- What about mental disorders? Were such diagnosed and were such patients included?
11. Figure 2. - Please provide better quality graphics. Please confirm if this is a graphic created by the authors?
12. L148 –‘’ E-mail: info@noraxon.com’’ - This information is not necessary.
13. L151 - ‘’(Elabd et al., 2020)’’ ’’ - Standardise the citation style.
14. L151-155 - The authors describe this rather enigmatically please insert a graphic with the location of the electrodes. Was this in accordance with the SENIAM programme?
15. In addition, information such as electrode impedance, manufacturability, form of signal processing etc. is missing. An article on sEMG reproduction has been published. Please refer to Table S1. - 10.3390/jcm13051328 - structure the description and answer the questions it contains:
‘’ 1. How was the skin prepared for the study? ž 1.1. What was used to cleanse the skin and how? ž 1.2. During what hours was the sEMG study conducted? ž 2. How were the electrodes positioned? ž 2.1. Was the placement in accordance with the SENIAM program guidelines? ž 2.2. What kind of electrodes were used (what was the conductive surface area, and the specific manufacturer)? ž 2.3. What was the maximum acceptable electrode impedance in the study? ž 3. What was the subject's position during the sEMG study? ž 4. What was the research procedure like? ž 4.1. What activities were performed by the subject during the study? ž 4.2. How many repetitions were performed (what were the intervals between repetitions)? ž 5. What sEMG equipment was used, and the specific manufacturer? ž 5.1. What was the sample rate? ž 5.2. What was the bandwidth (high-pass filter and low-pass filter cutoff frequencies)? ž 5.3. What is the input impedance of the used sEMG? ž 5.4. What was the common mode rejection ratio? ž 5.5. What was the input range? ž 5.6. What was the baseline noise? ž 5.7. What were the other important details about the device? ž 6. How was the signal processed? ž 6.1. Was there automatic processing? ž 6.1.1. If yes, was the exact name of the program and distributor given? ž 6.1.2. If no: 6.1.2.1. what filters were used? ž 6.1.2.2. in what order were the filters used? ž 6.1.2.3. what software was used for this purpose (name of the software and the specific manufacturer)?’’
Table S1. - 10.3390/jcm13051328
16. The authors write about EMG however, from the description I conclude that it is sEMG (superficial electromyography). Correct the abbreviation throughout the paper.
17. L206 –‘’ (Boolani et al., 2023).’’ - At this point, I stop pointing out errors relating to citation style. Please correct throughout the text.
18. L224 - add under paragraph Statistical analysis.
19. L231 – ‘’ (α) of 0.05’’ - add a multiplicity of effect to each result p. The value of p alone is not sufficient. The work can be helpful: doi: 10.4300/JGME-D-12-00156.1 and ‘’TRENDS in Sport Sciences 2014; 1(21): 19-25. ISSN 2299-9590’’
20. L231 - Why don't they use Bonferroni correction for p size?
21. L232- add sample size calculations.
22. Table 1. - What tests were used for these analyses?
23. Table 4 - The bottom of the table is missing.
24. Add sample size, Bonferroni correction, effect size, depending on new data improve discussion and conclusions. Until improved, I withdraw from the evaluation of discussion and conclusions.
25. Correct references according to the citation style of the journal.
26. L435-446 - Remove this information.
Best regards
Author Response
Attached is my response to your comments

Reviewer 2 Report
Comments and Suggestions for Authors 1. Please include more references in the Introduction. There are no references after line 74. 2. Please remove content after stating the research objectives. 3. For instance, provide a more detailed comparison between references 5-8 and the present study. 4. Add annotations to tables and figures to explain abbreviations and other details. It is recommended to separate them from the main text. 5. Please add units to Table 1. 6. In the Discussion, although previous studies and the this study's results are reported to be consistent or similar, explain the reasons and causes by citing previous studies.7. Line 309 consists of one paragraph that forms a single sentence. Consider merging it with another paragraph or adding additional content.
The tables and figures in the manuscript are too close to the main text. It seems that more space is needed between them. Please consider deleting the example at the end of the references when submitting the final version.
Author Response
Dear Reviewer,
We appreciate you for your precious time spent reviewing our paper and providing valuable comments. It was your valuable and insightful comment that led to further improvements in the current submission. The authors have carefully considered the comments and tried their best to address every one of them. We hope the manuscript, after careful revision, meets the high standards for publication in this journal. The authors welcome further constructive comment, if any. Below, we provide the point-by-point responses.
- Please include more references in the Introduction. There are no references after line 74.
Some references were added to the introduction
- Please remove content after stating the research objectives.
removed
- For instance, provide a more detailed comparison between references 5-8 and the present study.
All the references used the machine learning models for predicting diseases or subjective information but in different aspects of the science
- Add annotations to tables and figures to explain abbreviations and other details. It is recommended to separate them from the main text.
According to MDPI template provided to me, I placed them inside text as requested
- Please add units to Table 1.
added units
- In the Discussion, although previous studies and the this study's results are reported to be consistent or similar, explain the reasons and causes by citing previous studies.
Explained
- Line 309 consists of one paragraph that forms a single sentence. Consider merging it with another paragraph or adding additional content.
Combined with the previous paragraph
The tables and figures in the manuscript are too close to the main text. It seems that more space is needed between them. Please consider deleting the example at the end of the references when submitting the final version.
Spaces were left before and after the figures and tables
Round 2
Reviewer 1 Report
Comments and Suggestions for Authors
Here are my consecutive comments:
L116 – 'surface electromyographic (sEMG) parameters' – you have already expanded the abbreviation once. Please use the abbreviation.
L249-250 – 'Why don't they use Bonferroni correction for p size?' – 'Bonferroni’s posthoc'. 'Bonferroni correction' is not the same as 'Bonferroni’s posthoc'. Correction involves reducing the p-value. Divide the p-value by the number of groups. Please correct this.
Table 4: - add a bottom border to the table.
L395 – '..' remove the unnecessary period at the end of the sentence.
Add periods at the end of sentences in L402 and L405.
I accept the authors' remaining responses.
Author Response
We would like to express our gratitude for the opportunity to revise our manuscript based on the insightful comments provided. We have thoroughly addressed each comment and believe that the revisions have significantly improved the manuscript.
Please find below a summary of our responses to the comments:
L116 – 'surface electromyographic (sEMG) parameters' – you have already expanded the abbreviation once. Please use the abbreviation.
Done
L249-250 – 'Why don't they use Bonferroni correction for p size?' – 'Bonferroni’s posthoc'. 'Bonferroni correction' is not the same as 'Bonferroni’s posthoc'. Correction involves reducing the p-value. Divide the p-value by the number of groups. Please correct this.
Even with using berferoni correction the adjusted significance level would be 0.05/6 = 0.0083. all p-values listed as "< 0.001" and "0.002" in our results remain well below the adjusted significance threshold of 0.0083.
Table 4: - add a bottom border to the table.
Corrected
L395 – '..' remove the unnecessary period at the end of the sentence.
Corrected
Add periods at the end of sentences in L402 and L405.
Added
Also, please find the attached version of the manuscript with the changes highlighted
